# MetaReg: Towards Domain Generalization using Meta-Regularization

**Yogesh Balaji**
Department of Computer Science
University of Maryland
College Park, MD
yogesh@cs.umd.edu

**Swami Sankaranarayanan**[*]
Butterfly Network Inc.
New York, NY
swamiviv@butterflynetinc.com

**Rama Chellappa**
Department of Electrical and Computer Engineering
University of Maryland
College Park, MD
rama@umiacs.umd.edu

## Abstract

Training models that generalize to new domains at test time is a problem of fundamental importance in machine learning. In this work, we encode this notion of domain generalization using a novel regularization function. We pose the problem of finding such a regularization function in a Learning to Learn (or) meta-learning framework. The objective of domain generalization is explicitly modeled by learning a regularizer that makes the model trained on one domain to perform well on another domain. Experimental validations on computer vision and natural language datasets indicate that our method can learn regularizers that achieve good cross-domain generalization.

## 1 Introduction

Existing machine learning algorithms including deep neural networks achieve good performance in cases where the training and the test data are sampled from the same distribution. While this is a reasonable assumption to make, it might not hold true in practice. Deploying the perception system of an autonomous vehicle in new environments compared to its training setting might lead to failure owing to the shift in data distribution. Even strong learners such as deep neural networks are known to be sensitive to such domain shifts [9][33]. Approaches that resolve this issue in a domain adaptation framework have access to the target distribution. This is hardly true in practice: deploying real systems involve generalizing to unseen sources of data. This problem, also known as *domain generalization* is the focus of this paper.

Most machine learning models (including neural networks) are susceptible to domain shift - models trained on one dataset perform poorly on a related but a shifted dataset. Approaches for addressing this issue can be broadly grouped into two major categories - domain adaptation and domain generalization. Domain adaptation techniques assume access to target dataset, the models are trained using a combination of labeled source data and unlabeled/ sparsely labeled target data. Domain generalization, on the other hand, is a much harder problem as we assume no access to target information. Instead, the variations in multiple source domains are utilized to generalize to novel test distributions.

---

[*]Work done while at University of Maryland, College Park.

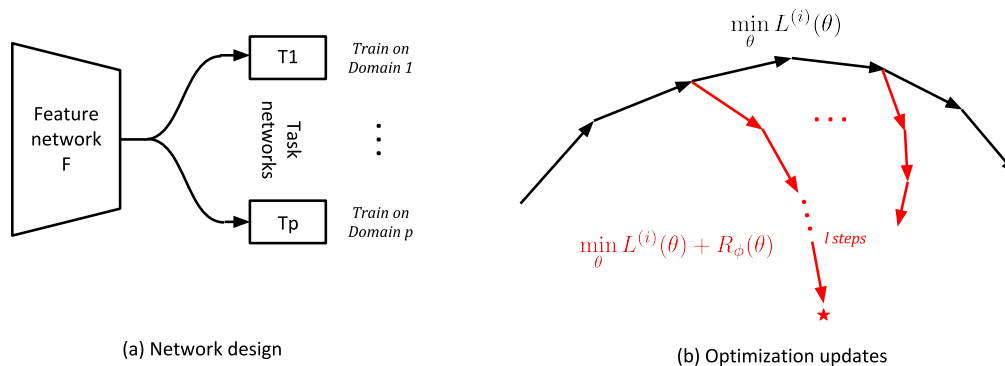

(a) Network design

(b) Optimization updates

Figure 1: Illustration of the proposed approach. Figure (a) depicts the network design - We employ a shared feature network $F$ and $p$ task networks $\{T_i\}_{i=1}^p$. Each task network $T_i$ is trained only on the data from domain $i$, and the shared network $F$ is trained on all $p$ source domains. The figure on the right illustrates the optimization updates. At each iteration we sample a pair of domains $(i, j)$ from the training set. The black arrows are the SGD updates of the task network $T_i$ trained on domain $i$. From each point in the black path, we take $l$ gradient steps using the regularized loss and the samples from domain $i$ to reach a new point $*$. We then compute the loss on domain $j$ at $*$. The regularizer parameters $\phi$ are updated so that this meta-loss is minimized. This ensures that the task network $T_i$ trained with the proposed regularizer generalizes to domain $j$

The conventional approach to improve the generalization of a parametric model is to introduce regularization in the loss function [32]. In a Bayesian setting, regularization can be interpreted as priors on the parameters. Several regularization schemes have been proposed for neural networks including weight decay [17], Dropout [28], DropConnect [31], batch normalization [14], etc. While these schemes have been shown to reduce test error on samples drawn from the same training distribution, they do not generalize when there is a training-test distribution mismatch. Hence, the objective of this work is to learn a regularizer that generalizes to novel distributions not present in the training data.

Designing such regularizers for achieving cross-domain generalization is a challenging problem. The difficulty in mathematically modeling domain shifts makes it hard to design hand-crafted regularizers. Instead, we take a data-driven approach where we aim to learn the regularization function using the variability in the source domains. We cast the problem of learning regularizers in a *learning to learn*, or *meta-learning* framework, which has received a resurgence in interest recently with applications including few-shot learning [7][24] and learning optimizers [20][1]. Similar to [7], we follow an episodic paradigm where at each iteration, we sample an episode comprising meta-train and meta-test data such that the domains contained in meta-train and meta-test sets are disjoint. The objective is then to train the regularizer such that $k$ steps of gradient descent using the meta-train data results in decreasing the loss in the meta-test. This procedure is repeated for multiple episodes sampled from the source dataset. After the regularizer is trained, we fine-tune a new model on the entire source dataset using the trained regularizer.

The primary contribution of this work is that we propose a scheme for learning regularization functions that enable domain generalization. We show how the notion of domain generalization can be explicitly encoded in a regularization function, which can then be used to train models that are more robust to domain shifts. This framework is also scalable as the same regularizer can later be used to fine-tune on a larger dataset. Experiments indicate that our approach can learn regularizers that achieve good cross-domain generalization on benchmark domain generalization datasets.

## 2 Related work

**Domain Adaptation**   Domain adaptation has received significant attention in recent years from machine learning, computer vision and natural language processing communities. Non-deep learning

approaches to this problem include feature engineering methods [6], learning intermediate subspaces using manifolds [11][12] and dictionaries [23], etc. Recent methods harness the expressive power of deep neural networks to learn domain invariant representations. The method in [9] attempts to reduce the distributional distance between source and target embeddings by formulating a minimax game between the feature network and a domain discriminator network. [5] uses stacked denoising autoencoders for learning robust data representations that adapt well to target domains. Other notable works include the use of Maximum Mean Discrepancy (MMD) [21], Generative Adversarial Networks (GAN) [25][26], co-training [4], etc.

**Meta-learning** The concept of meta-learning (or) learning to learn has a long standing history, some of the earlier works include [30] [27]. Recently, there has been a lot of interest in applying such strategies for deep neural networks. One interesting application is the problem of learning the optimization updates of neural networks by casting it as a policy learning problem in a Markov decision process [1] [20]. Few-shot learning is another problem where meta-learning strategies have been widely explored. [24] proposes an LSTM-based meta learner for learning the optimization updates of a few-shot classifier. Instead of learning the updates, [7] learns transferable weight representations that quickly adapts to a new task using only a few samples. Other recent applications that use meta learning include imitation learning [8], visual question answering [29], etc.

**Domain Generalization** Unlike domain adaptation, domain generalization is a relatively less explored area of research. [22] proposes domain invariant component analysis, a kernel-based algorithm for minimizing the differences in the marginal distributions of multiple domains. [10] attempts to learn a domain-invariant feature representation by using multi-view autoencoders to perform cross domain reconstructions. The method in [15] decomposes the parameters of a model (SVM classifier) into domain specific and domain invariant components, and uses the domain invariant parameters to make predictions on the unseen domain. [18] extends this idea to decompose the weights of deep neural networks using multi-linear model and tensor decomposition.

Finn *et al.* [7] recently proposed a model agnostic meta-learning procedure for few shot learning problems. The objective of the MAML approach is to find a good initialization $\theta$ such that few gradient steps from $\theta$ results in a good task specific network. The focus of the MAML is to adapt quickly in few shot settings. Recently, [19] proposed a meta learning based approach (MLDG) extending MAML to the domain generalization problem. This approach has the following limitations - the objective function of MAML is more suited for fast task adaptation for which it was originally proposed. In domain generalization however, we do not have access to samples from a new domain, and so a MAML-like objective might not be effective. The second issue is scalability - it is hard to scale MLDG to deep architectures like Resnet [13]. Our approach attempts to tackle both these problems - (1) We explicitly address the notion of domain generalization in our episodic training procedure by using a regularizer to go from a task specific representation to a task general representation at each episode. (2) We make our approach scalable by freezing the feature network and performing meta learning only on the task network. This enables us to use our approach to train deeper models like Resnet-50. A similar approach for training meta-learning algorithms in feature space has been explored in a recent work [34].

## 3 Method

### 3.1 Problem Setup

We begin with a formal description of the domain generalization problem. Let $\mathcal{X}$ denote the instance space (which can be images, text, etc.) and $\mathcal{Y}$ denote the label space. Domain generalization involves data sampled from $p$ source distributions and $q$ target distributions, each containing data for performing the same task. Classification tasks are considered in this work. Hence, $\mathcal{Y}$ is the discrete set $\{1, 2, \ldots N_c\}$, where $N_c$ denotes the number of classes. Let $\{\mathcal{D}_i\}_{i=1}^{p+q}$ represent the $p + q$ distributions, each of which exists on the joint space $\mathcal{X} \times \mathcal{Y}$. Let $D_i = \{(\mathbf{x}_j^{(i)}, y_j^{(i)})\}_{j=1}^{N_i}$ represent the dataset sampled from the $i^{th}$ distribution, i.e. each $(x_j^i, y_j^i) \overset{i.i.d.}{\sim} \mathcal{D}_i$. In the rest of the paper, $D_i$ is referred to as the $i^{th}$ *domain*. Note that every $D_i$ shares the same label space. In the domain generalization problem, each of the $p + q$ domains contain varied domain statistics. The objective is to train models on the $p$ source domains so that they generalize well to the $q$ novel target domains.

We are interested in training a parametric model $M_\Theta : \mathcal{X} \to \mathcal{Y}$ using data only from the $p$ source domains. In this work, we consider $M_\Theta$ to be a deep neural network. We decompose the network $M$ into a feature network $F$ and a task network $T$ (i.e) $M_\Theta(\mathbf{x}) = (T_\theta \circ F_\psi)(\mathbf{x})$, where $\Theta = \{\psi, \theta\}$. Here, $\psi$ denotes the weights of the feature network $F$, and $\theta$ denotes the weights of the task network. The output of $M_\Theta(\mathbf{x})$ is a vector of dimension $N_c$ with $i^{th}$ entry denoting the probability that the instance $\mathbf{x}$ belongs to the class $i$. Standard neural network training involves minimizing the cross entropy loss function given by Eq (1)

$$L(\psi, \theta) = \mathbb{E}_{(\mathbf{x}, y) \sim D}[-\mathbf{y} . \log(M_\Theta(\mathbf{x}))] = \sum_{i=1}^{p} \sum_{j=1}^{N_i} -\mathbf{y}_j^{(i)} . \log(M_\Theta(\mathbf{x}_j^{(i)})) \qquad (1)$$

Here, $\mathbf{y}_j^{(i)}$ is the one-hot representation of the label $y_j^{(i)}$ and '.' denotes the dot product between two vectors. The above loss function does not take into account any factor that models domain shifts, so generalization to a new domain is not expected. To accomplish this, we propose using a regularizer $R(\psi, \theta)$. The new loss function then becomes $L_{reg}(\psi, \theta) = L(\psi, \theta) + R(\psi, \theta)$. The regularizer $R(\psi, \theta)$ should capture the notion of domain generalization (i.e) it should enable generalization to a new distribution with varied domain statistics. Designing such regularizers is hard in general, so we propose to learn it using meta learning.

## 3.2 Learning the regularizer

In this work, we model the regularizer $R$ as a neural network parametrized by weights $\phi$. Moreover, the regularization is applied only on the parameters $\theta$ of the task network to enable scalable meta-learning. So, the regularizer is denoted as $R_\phi(\theta)$ in the rest of the paper. We now discuss how the parameters of the regularizer $R_\phi(\theta)$ are estimated. In this stage of the training pipeline, the neural network architecture consists of a feature network $F$ and $p$ task networks $\{T_i\}_{i=1}^{p}$ (with parameters of $T_i$ denoted by $\theta_i$) as shown in Fig. 1. Each $T_i$ is trained only on the samples from domain $i$ and $F$ is the shared network trained on all $p$ source domains. The reason for using $p$ task networks is to enforce domain-specificity in the models so that the regularizer can be trained to make them domain-invariant.

We now describe the procedure for learning the regularizer:

- **Base model updates:** We begin by training the shared network $F$ and $p$ task networks $\{T_i\}_{i=1}^{p}$ using supervised classification loss $L(\psi, \theta)$ given by Eq (1). Note that there is no regularization in this step. Let the network parameters at the $k^{th}$ step of this optimization be denoted as $[\psi^{(k)}, \theta_1^{(k)}, \ldots \theta_p^{(k)}]$.

- **Episode creation:** To train $R_\phi(\theta)$, we follow an episodic training procedure similar to [19]. Let $a, b$ be two randomly chosen domains from the training set. Each episode contains data partitioned into two subsets - (1) $m_1$ labeled samples from domain $a$ denoted as *metatrain* set and (2) $m_2$ labeled samples from domain $b$ denoted as *metatest* set. The domains contained in both the sets are disjoint (i.e) $a \neq b$, and the data is sampled only from the source distributions (i.e) $a, b \in \{1, 2, \ldots p\}$.

- **Regularizer updates** At iteration $k$, a new task network $T_{new}$ is initialized with $\theta_a^{(k)}$ - the base model's task network parameters of the $a^{th}$ domain at iteration $k$. Using the samples from the *metatrain* set (which contains domain $a$), $l$ steps of gradient descent is performed with the regularized loss function $L_{reg}(\psi, \theta)$ on $T_{new}$. Let $\hat{\theta}_a^{(k)}$ denote the parameters of $T_{new}$ after these $l$ gradient steps. We treat each update of the network $T_{new}$ as a separate variable in the computational graph. $\hat{\theta}_a^{(k)}$ then depends on $\phi$ through these $l$ gradient steps. The unregularized loss on the *metatest* set computed using $T_{new}$ (with parameters $\hat{\theta}_a^{(k)}$) is then minimized with respect to the regularizer parameters $\phi$. Each regularizer update unrolls through the $l$ gradient steps as $\hat{\theta}_a^{(k)}$ depends on $\phi$ through the $l$ gradient steps. This entire procedure can be expressed by the following set of equations:

$$\beta^1 \leftarrow \theta_a^{(k)}$$

$$\beta^t = \beta^{t-1} - \alpha \nabla_{\beta^{t-1}} \left[ L^{(a)}(\psi^{(k)}, \beta^{t-1}) + R_\phi(\beta^{t-1}) \right] \qquad \forall t \in \{2, \dots l\} \quad (2)$$

$$\hat{\theta}_a^{(k)} = \beta^l$$

$$\phi^{(k+1)} = \phi^{(k)} - \alpha \nabla_\phi L^{(b)}(\psi^{(k)}, \hat{\theta}_a^{(k)})|_{\phi=\phi^{(k)}} \tag{3}$$

Here, $L^{(i)}(\psi, \theta_{new}) = \mathbb{E}_{(\mathbf{x},\mathbf{y}) \sim D_i}[-\mathbf{y}. \log(T_{\theta_{new}}(F_\psi(\mathbf{x})))]$ (i.e) the loss of task network $T_{new}$ on samples from domain $i$, and $\alpha$ is the learning rate. Eq (2) represents $l$ steps of gradient descent from the initial point $\theta_a^{(k)}$ using samples from *metatrain* set, with $\beta_t$ denoting the output at the $t^{th}$ step. Eq (3) is the meta-update step for updating the parameters of the regularizer. This update ensures that $l$ steps of gradient descent using the regularized loss on samples from domain $a$ results in task network $a$ performing well on domain $b$.

It is important to note that the dependence of $\phi$ on $\hat{\theta}_a^{(k)}$ comes from the $l$ gradient steps performed in Eq. 2. So, the gradients of $\phi$ propagates through these $l$ unrolled gradient steps.

Since the same regularizer $R_\phi(\theta)$ is trained on every $(a, b)$ pair, the resulting regularizer we learn captures the notion of domain generalization. Please refer to Fig. 1 for a pictoral description of the meta-update step. The entire algorithm is given in Algorithm 1

### 3.3 Training the final model

Once the regularizer is learnt, the regularization parameters $\phi$ are frozen and the final task network initialized from scratch is trained on all $p$ source domains using the regularized loss function $L_{reg}(\psi, \theta)$. The network architectures consists of just one $F - T$ pair. In this paper, we use weighted $L_1$ loss as our regularization function, (i.e) $R_\phi(\theta) = \sum_i \phi_i |\theta_i|$. The weights of this regularizer are estimated using the meta-learning procedure discussed above. However, our approach is general and can be extended to any class of regularizers (refer to Section. 5). The use of weighted $L_1$ loss can be interpreted as a learnable weight decay mechanism - Weights $\theta_i$ for which $\phi_i$ is positive will be decayed to 0 and those for which $\phi_i$ is negative will be boosted. By using our meta-learning procedure, we select a common set of weights that achieve good cross-domain generalization across every pair of source domains $(a, b)$.

---

**Algorithm 1** MetaReg training algorithm

---

**Require:** $N_{iter}$: number of training iterations
**Require:** $\alpha_1, \alpha_2$: Learning rate hyperparameters
1: **for** $t$ in $1 : N_{iter}$ **do**
2:     **for** $i$ in $1 : p$ **do**
3:         Sample $n_b$ labeled images $\{(x_j^{(i)}, y_j^{(i)}) \sim D_i\}_{j=1}^{n_b}$
4:         Perform supervised classification updates:
5:         $\psi^{(t)} \leftarrow \psi^{(t-1)} - \alpha_1 \nabla_\psi L^{(i)}(\psi^{(t-1)}, \theta_i^{(t-1)})$
6:         $\theta_i^{(t)} \leftarrow \theta_i^{(t-1)} - \alpha_1 \nabla_{\theta_i} L^{(i)}(\psi^{(t-1)}, \theta_i^{(t-1)})$
7:     **end for**
8:     Choose $a, b \in \{1, 2, \dots p\}$ randomly such that $a \neq b$
9:     $\beta^1 \leftarrow \theta_a^{(t)}$
10:     **for** $i = 2 : l$ **do**
11:         Sample *metatrain* set $\{(x_j^{(a)}, y_j^{(a)}) \sim D_a\}_{j=1}^{n_b}$
12:         $\beta^i = \beta^{i-1} - \alpha_2 \nabla_{\beta^{i-1}}[L^{(a)}(\psi^{(t)}, \beta^{i-1}) + R_\phi(\beta^{i-1})]$
13:     **end for**
14:     $\hat{\theta}_a^{(t)} = \beta_l$
15:     Sample *metatest* set $\{(x_j^{(b)}, y_j^{(b)}) \sim D_b\}_{j=1}^{n_b}$
16:     Perform meta-update for regularizer $\phi^{(t)} = \phi^{(t-1)} - \alpha_2 \nabla_\phi L^{(b)}(\psi^{(t)}, \hat{\theta}_a^{(t)})|_{\phi=\phi^{(t)}}$
17: **end for**

---

Table 1: Cross-domain recognition accuracy (in %) averaged over 5 runs on PACS dataset using Alexnet architecture. For the baseline setting, the numbers on the parenthesis indicate the baseline performance as reported by [19]

| Method | Art painting | Cartoon | Photo | Sketch | Average |
|---|---|---|---|---|---|
| Baseline | $67.21 \pm 0.72$ (64.91) | $66.12 \pm 0.51$ (64.28) | $88.47 \pm 0.63$ (86.67) | $55.32 \pm 0.44$ (53.08) | 69.28 (67.24) |
| D-MTAE ([10]) | 60.27 | 58.65 | **91.12** | 47.68 | 64.48 |
| DSN ([3] | 61.13 | 66.54 | 83.25 | 58.58 | 67.37 |
| DBA-DG ([18]) | 62.86 | 66.97 | 89.50 | 57.51 | 69.21 |
| MLDG ([19]) | 66.23 | 66.88 | 88.0 | 58.96 | 70.01 |
| MetaReg (*Ours*) | **69.82** $\pm 0.76$ | **70.35** $\pm 0.63$ | $91.07 \pm 0.41$ | **59.26** $\pm 0.31$ | **72.62** |

### 3.4 Summary of the training pipeline

The feature network is first trained using combined data from all source domains, and is kept frozen in the rest of training. The regularizer parameters are then estimated using the meta-learning procedure described in the previous section. As the individual task networks are updated on their respective source domain data, the regularizer updates are derived from each point of this SGD path with the objective of cross-domain generalization (refer Alg. 1). To learn the regularizer effectively at the early stages of the task network updates, replay memory is used where the regularizer updates are periodically derived from the early stages of the task networks' SGD paths. The learnt regularizer is used in the final step of the training process where a single $F - T$ network is trained using the regularized cross-entropy loss.

## 4 Experiments

In this section, we describe the experimental validation of our proposed approach. We perform experiments on two benchmark domain generalization datasets - Multi-domain image recognition using PACS dataset [18] and sentiment classification using Amazon Reviews dataset [2].

### 4.1 PACS dataset

PACS dataset is a recently proposed benchmark dataset for domain generalization. This dataset contains images from four domains - *Photo*, *Art painting*, *Cartoon* and *Sketch*. Following [19], we perform experiments on four settings: In each setting, one of the four domains is treated as the unseen target domain, and the model is trained on the other three source domains.

**Alexnet** The first set of experiments is based on the Alexnet [16] model pretrained on Imagenet. The feature network $F$ comprises of the top layers of Alexnet model till $pool5$ layer, while the task network $T$ contains $fc6$, $fc7$ and $fc8$ layers. For the regularizer network, we used weighted $L_1$ loss (i.e) $R_\phi(\theta) = \sum_i \phi_i |\theta_i|$, where $\phi_i$ are the parameters estimated using meta-learning. In all our experiments, $Baseline$ setting denotes training a neural network (Alexnet in this case) on all of the source domains without performing any domain generalization. Other comparison methods include Multi-task Autoencoders (MTAE) [10], Domain Separation Networks (DSN) [3], Artier Domain Generalization (DBA-DG) [18] and MLDG [19]. While some of these methods were originally proposed for domain adaptation, they were adapted to the domain generalization problem as done in [19].

All our models are trained using the SGD optimizer with learning rate $5e - 4$ and a batch size of $64$. This is in accordance with the setup used in [19]. Table 1 presents the results of our approach along with other comparison methods. We observe that our method obtains a performance improvement of $3.34\%$ over the baseline, thus achieving the state-of-the-art performance on this dataset.

**Resnet** One disadvantage with approaches like MLDG [19] is that it requires differentiating through $k$ steps of optimization updates, and this might not be scalable to deeper architectures like Resnet. Even our approach requires a similar optimization process. However, unlike [19], we perform meta-learning only on the task network. Since the task network is much shallower than the feature network, our approach is scalable even to some of the contemporary deep architectures. In this section, we show experiments using two such architectures - Resnet18 and Resnet 50.

We use the Resnet-18 and Resnet-50 models pretrained on ImageNet as our feature network, and the last fully connected layer as our task network. Similar to the previous experiment, we used weighted $L_1$ loss as our class of regularizers. All models were trained using SGD optimizer with a learning rate of 0.001 and momentum 0.9. The hyper-parameters $\alpha_1$ and $\alpha_2$ are both set as 0.001. The results of our experiments are reported in Table. 2. Our method performs better than baseline in both settings. It is important to note that the baseline numbers for Resnet architectures are much higher than that of Alexnet. Even on such stronger baselines, our method gives performance improvement.

## 4.2 Sentiment Classification

In this section, we perform experiments on the task of sentiment classification on *Amazon reviews* dataset as pre-processed by [5]. The dataset contains reviews of products belonging to four domains - *books*, *DVD*, *electronics* and *kitchen appliances*. The differences in textual description of the reviews each of these product categories manifests as domain shift. Following [9], we use unigrams and bigrams as features resulting in 5000 dimensional vector representations. The reviews are assigned binary labels - 0 if the rating of the product is upto 3 stars, and 1 if the rating is 4 or 5 stars.

We conduct 4 cross-domain experiments - in each setting one of the four domains is treated as the unseen test domain, and the other three domains are used as source domains. Similar to [9], we used a neural network with one hidden layer (with 100 neurons) as our task network. All models were trained using an SGD optimizer with learning rate 0.01 and momentum 0.9 for 5000 iterations. The results of our experiments are reported in the Table. 3. Since there is significant variation in performance over runs, each experiment was repeated 10 times with different random weight initialization and averages of these 10 runs are reported. We observe that our method performs better than the baseline in all of the settings. However, the performance improvement is less compared to the previous experiments. This is because of the nature of the problem and the architectural choice. We would like to point out that even domain adaptation methods that make use of unlabeled target data achieve similar gains in performance [9] in this dataset.

## 5 Ablation Study

For all the ablation experiments except 5.3, we use the Resnet-18 model as our neural network architecture, and Art-painting setting in PACS dataset as our experimental setting, (i.e) we use *Art painting* domain as the test domain, and *Cartoon*, *Photo* and *Sketch* as source domains.

### 5.1 Class of Regularizers

In this experiment, we study the effect of different regularizers on the performance of our approach. We experimented on the following class of regularizers: (1) Weighted $L_1$ loss: $R_\phi(\theta) = \sum_i \phi_i |\theta_i|$, (2) Weighted $L_2$ loss: $R_\phi(\theta) = \sum_i \phi_i \theta_i^2$, and (3) Two layer neural network: $R_\phi(\theta) = \phi^{(2)T}(ReLU(\phi^{(1)T}\theta))$. The performance of these regularizers are reported in Table. 4. We observe that the Weighted $L_1$ regularizer performs the best among the three. Also, we observed that training networks with the weighted $L_1$ regularizer lead to better convergence and stability in performance compared to the other two. We also compare our approach with two other schemes: (1) DropConnect [31] and (2) Default $L_1$ regularization, which is Weighted $L_1$ regularization where the weights $\phi_i = 1$. We observe that both these schemes do not improve the baseline performance.

Table 2: Cross-domain recognition accuracy (in %) averaged over 5 runs on PACS dataset using Resnet architectures

| Method | Art painting | Cartoon | Photo | Sketch | Average |
|---|---|---|---|---|---|
| Resnet-18 | | | | | |
| Baseline | $79.9 \pm 0.22$ | $75.1 \pm 0.35$ | $95.2 \pm 0.18$ | $69.5 \pm 0.37$ | 79.9 |
| Metareg (*Ours*) | $\textbf{83.7} \pm 0.19$ | $\textbf{77.2} \pm 0.31$ | $\textbf{95.5} \pm 0.24$ | $\textbf{70.3} \pm 0.28$ | **81.7** |
| Resnet-50 | | | | | |
| Baseline | $85.4 \pm 0.24$ | $77.7 \pm 0.31$ | $\textbf{97.8} \pm 0.17$ | $69.5 \pm 0.42$ | 82.6 |
| Metareg (*Ours*) | $\textbf{87.2} \pm 0.13$ | $\textbf{79.2} \pm 0.27$ | $97.6 \pm 0.31$ | $\textbf{70.3} \pm 0.18$ | **83.6** |

Table 3: Cross domain classification accuracy (x %) averaged over 10 runs on Amazon Reviews dataset

| Method | Books | DVD | Electronics | Kitchen | Average |
|---|---|---|---|---|---|
| Baseline | $75.5 \pm 0.52$ | $79.0 \pm 0.37$ | $83.7 \pm 0.44$ | $84.7 \pm 0.63$ | 80.7 |
| Metareg (*Ours*) | $\mathbf{76.1} \pm 0.41$ | $\mathbf{79.6} \pm 0.32$ | $\mathbf{83.9} \pm 0.28$ | $\mathbf{85.1} \pm 0.43$ | **81.2** |

Table 4: Effect of different classes of regularization functions

| Baseline | DropConnect [31] | Default $L_1$ | Weighted $L_1$ | Weighted $L_2$ | 2 layer NN |
|---|---|---|---|---|---|
| 79.9 | 80.1 | 79.7 | 83.7 | 83.2 | 83.3 |

## 5.2 Availability of data over time

In all of the previous experiments, we assumed that the entire training data is available from the start of the training process. But consider a more general setting where we train our model on some initial data, but more data gets available over time. Is it possible make use of the newly available data to improve our models without having to perform meta-learning again? We propose the following solution: Train the feature network, task network and regularizer on the initial dataset. On the new data, finetune the task network and feature network using the regularizer trained on the initial data. Note that, we do not perform meta learning again on the new data, and so this is computationally efficient since meta-learning procedure incurs significant overhead over a regular finetuning process. With approaches like MLDG [19], meta-learning has to be performed even on the new data.

We simulate these experimental conditions as follows: In each setting, we consider a fraction $f$ of the PACS dataset as our intial dataset on which our model and the regularizer are trained. We then finetune our model on the remaining data using the trained regularizer. The performance of these models on the test set are shown in Table. 5. We observe that there is little drop in performance for all $f$ values. Our approach is able to learn good regularizers even with $10\%$ of the entire dataset.

Table 5: Experiments for training models on less data

| Data fraction $f$ | 0.1 | 0.2 | 0.3 | 0.4 | 0.5 | 1.0 |
|---|---|---|---|---|---|---|
| *Accuracy (in %)* | 82.86 | 83.11 | 83.42 | 83.62 | 83.60 | 83.71 |

## 5.3 Effect of the number of layers regularized

In our training paradigm, the neural network is decomposed into feature and task network, and the meta-regularization is performed only on the task network. Deciding this feature/task network split is a design choice which needs to be understood. The effect of domain generalization performance on varying the number of layers is reported in Table 6. This experiment is performed using the Alexnet architecture on PACS dataset with *Cartoon* as the target domain. We observe that as the number

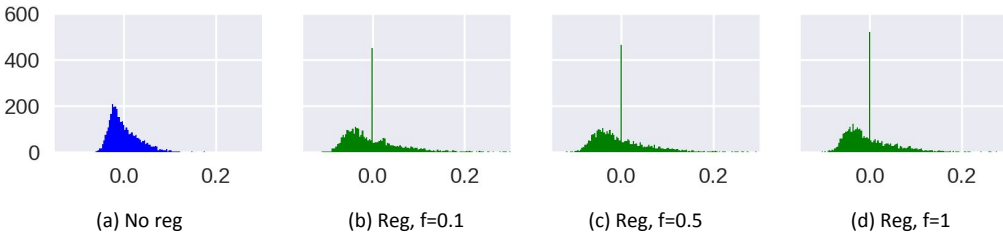

Figure 2: Histogram of the weights learnt by the task network. "No reg" corresponds to the network without regularizaton, and "Reg, f=x" corresponds to the regularized network, where the regualrizer $R$ is trained only on $x\%$ of the data

of regularization layers increases, the generalization performance increases and saturates beyond a point.

Table 6: Effect of cross-domain generalization with varying number of layers regularized on PACS dataset using Alexnet model. Cartoon is used as the test domain

| Layers regularized | None | $fc_8$ | $fc_7 + fc_8$ | $fc_6 + fc_7 + fc_8$ |
|---|---|---|---|---|
| *Accuracy (in %)* | 66.12 | 67.31 | 70.10 | 70.35 |

## 5.4 Visualizing the weights

We plot the histogram of the weights learnt by the task network with and without the use of our regularizer in Fig. 2. The following observations can be made: (1) For the network with regularization, there is a sharp peak at $0$. This is because the weights $\theta_i$ for which $\phi_i$ are positive are decayed to $0$. (2) The weights of the network with regularization has wider spread compared to the network without regularization. This is because the weights $\theta_i$ for which $\phi_i$ are negative are boosted, due to which certain weights have high values.

## 6 Conclusion and Future Work

In this work, we addressed the problem of domain generalization by using regularization. The task of finding the desired regularizer that captures the notion of domain generalization is modeled as a meta-learning problem. Experiments indicate that the learnt regularizers achieve good cross-domain generalization on the benchmark domain generalization datasets. Some avenues for future work include scalable meta-learning approaches for learning regularization functions over convolutional layers while preserving the spatial dependency between the channels, and extending our approach to deep reinforcement learning problems.

## 7 Acknowledgement

This reseach was supported by MURI from the Army Research Office under the Grant No. W911NF-17-1-0304. This is part of the collaboration between US DOD, UK MOD and UK Engineering and Physical Research Council (EPSRC) under the Multidisciplinary University Research Initiative.

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
