[Supplementary Material]

# Supplementary Material
# MetaReg: Towards Domain Generalization using Meta-Regularization

**Bayesian interpretation**

In this section, we provide a Bayesian interpretation to our MetaReg approach. Let $\theta_1, \theta_2, \ldots \theta_p$ represent the parameters of $p$ task networks where each $\theta_i$ is trained on the dataset $D_i$. The objective is to find a representation $\theta_g$ by using the $p$ task networks, that generalizes well to unseen domains. The posterior density of the parameters $\theta_g$ directly depends on the data $X$ and implicitly depends on the $k$ task networks $\{\theta_i\}_{i=1}^k$. We introduce a latent variable $\phi$ in order to explicitly capture this dependence:

$$P(\theta_g|X, \theta_1, \ldots, \theta_p) = \int P(\theta_g, \phi|X, \theta_1, \ldots, \theta_p)d\phi$$

$$= \int P(\theta_g|X, \phi, \theta_1, \ldots \theta_p)P(\phi|X, \theta_1, \ldots \theta_p)d\phi$$

$$= \int P(\theta_g|X, \phi)P(\phi|X, \theta_1, \ldots \theta_p)d\phi \quad (1)$$

The last equation is the result of assuming that the domain specific parameters ($\{\theta_i\}_{i=1}^k$) and the domain general parameters ($\theta_g$) are conditionally independent given the latent variables $\phi$. Instead of integrating Eq. 1 over $\phi$ which is intractable, we make an approximation that uses a point estimate $\hat{\phi}_{meta}$. This point estimate is obtained via the meta-regularization approach described in Section 3 of the main paper, hence avoiding the need to perform integration over $\phi$. Minimizing the negative logarithm of the posterior density can then be written as:

$$\min_{\theta_g} -\log P(\theta_g|X, \theta_1, \ldots \theta_p) \approx \min_{\theta_g} -\log P(\theta_g|X, \hat{\phi}_{meta})$$

$$= \min_{\theta_g} -\log P(X|\theta_g) - \log P(\theta_g|\hat{\phi}_{meta}) \quad (2)$$

where the last equality results from a direct application of Bayes' rule. So, finding the domain general parameters $\theta_g$ involves a two step training procedure: (1) Obtaining the point estimate $\hat{\phi}_{meta}$ using meta learning and (2) Optimizing Eq. (2) using the learned regularizer parameters $\hat{\phi}_{meta}$.

In our analysis, we made an approximation of replacing $\phi$ by the point estimate $\phi_{meta}$( the parameters estimated using our meta-learning approach). The closeness of this approximation is a topic of future research.

**Effect of cross-domain generalization on the number of unrolling steps**

In this experiment, we examine the effect of number of unrolling SGD updates in the meta learning process (effect of $l$) on the cross-domain generalization performance. We performed experiments with $l = 1, 2, 3, 4, 5$. The results are reported in Table 1. Even with a 1- step update, our method achieves good performance improvement compared to the baseline. Performance keeps increasing with increasing $l$ and saturates after $l = 3$.

Table 1: Effect of number of unrolling steps

| # steps $l$ | Accuracy (in %) |
|---|---|
| 1 | 83.41 |
| 2 | 83.57 |
| 3 | 83.71 |
| 4 | 83.66 |
| 5 | 83.72 |

## All-but-one training

In the proposed framework, $p$ task networks $\{T_i\}_{i=1}^p$ are used for training the meta-regularizer. Each $T_i$ is trained only on domain $i$, and the regularizer is trained with the objective that $T_i$ must decrease loss on domain $j$ $\forall(i,j)$. Instead of training $T_i$ only on domain $i$, an alternative is to train $T_i$ on all domains other than $i$. The meta-regularizer can then be trained so that every $T_i$ decreases loss on domain $i$. This framework better mimics the domain generalization problem setup. In the two datasets we considered in this paper (PACS and Sentiment classification), this all-but-training did not gain any improvements over the training procedure in Algorithm 1 of the main paper. However, when the number of domains are large, this might lead to more performance gains.

## When does MetaReg work?

Understanding failure cases is important as it provides better insight on the workings of our approach. We study this on Rotated-MNIST dataset – dataset with MNIST digits rotated by $0°, 10°, 20°, 30°$ and $60°$, each of which correponds to one domain. The benefit of using this controlled dataset is that it is easier to quantify domain shifts. For instance, $10°$ rotations are closer to $0°$ than $75°$. In our experimets, the datasets corresponding to $0°$, $10°$ and $30°$ were used as source domains, and $20°$ and $60°$ rotations are used as target domains. A 2-layer MLP ($784 \rightarrow 128 \rightarrow 128 \rightarrow 10$) is used as the task network, no feature network was used. So, the entire network is regularized. Table 2 presents the results of the cross-domain generalization on these two target domains.

Table 2: Effect of cross-domain generalization on the extent of domain shift

| Method | Accuracy (in %) on $20°$ domain | Accuracy (in %) on $60°$ domain |
|---|---|---|
| Baseline | 95.9 | **57.3** |
| MetaReg | **96.7** | 56.8 |

We observe that there is an improvement in the performance on the $20°$ domain and a drop in performance on the $60°$ domain. This is because the $60°$ domain presents much larger domain shift than the variations represented in the training set. This suggests that MetaReg works as long as the shifts encountered in the test set is refective of the variations captured in the training domains.

## Tuning hyperparameters

One important aspect of *domain generalization* and *domain adaptation* problems is the tuning of hyperparameters. We need to tune our models without making use of the labeled samples in the target domain. In all our experiments, we adopt the following cross-validation strategy: From the $p$ source domains, we choose a domain $i$ as our target domain for cross-validation. We train our algorithm on the remaining $p-1$ source domains and test it on the $i^{th}$ domain. This is reperated for every $i$ in $\{1, 2, \ldots p\}$. The hyper-parameter setting that gives the best average cross-validation accuracy is then picked.