[Reviews · NeurIPS 2018]

Reviewer 1



MetaReg in a nutshell: This paper reinterprets and further develops few-shot meta-learning ideas for the challenging domain generalization paradigm, using standard supervised benchmarks. The main contribution is the learning of a regularizer, as opposed to learning an initial set of parameters well “positioned” for finetuning. Scores seem to be significantly improved in several cases, but I am not an expert. Pros: - The paper goes beyond the original inspiration and adapts the approach to serve a substantially different problem. While in meta-learning the degree of similarity between problem instances is substantial, e.g. random subset of classes from ImageNet, dealing well with a multi-modal meta-training set, due to differences between source domains is interesting in itself. - Several practical issues are resolved such that the approach is less expensive and more accurate than baselines. - Several baselines are considered, but I cannot say for sure how strong they are since I am less familiar with the state-of-the-art. - Analysis shows graceful degradation with reduced amounts of data, which is particularly impressive. Cons: - A substantial bit of the approach could be viewed as derivative if not enough explanation is given for why the miniImageNet task (for example) is not a good paradigm for testing “domain generalization” as precisely defined in the paper, at the conceptual level. - Furthermore, even if the paradigms are clearly different conceptually, say by definition, it would be good to discuss at length if these conceptual differences are really well embodied by the benchmarks used in the two cases. If the answer is negative, then proposing improved benchmarks in both cases would be useful. - Is learning a regularizer better than MAML because the assumptions of MAML are broken by this different paradigm and the benchmark is sensitive enough to show this? Or simply applying MAML with the same tricks and care would yield similar results? For example MAML is model-agnostic, so it could very well be applied only to task-specific networks. Furthermore, MAML but with meta-optimized per-parameter learning rates (Meta-SGD) could also be argued to be (implicitly) learning to regularize. Explanation of overall score: Quality: Well balanced paper with substantial experiments and some detailed analysis. Clarity: Haven’t had any difficulties. Originality: Ultimately I find it difficult to judge. Significance: Good scores on supervised benchmarks in a challenging paradigm are fine for research in that sub-field. It is not trivial to relate these results to few-shot meta-learning tasks.

Reviewer 2



** Post Rebuttal ** The rebuttal adequately addresses all of my suggestions and questions. All of the new experiments in the rebuttal are valuable and should be included in the final. I recommend accepting the paper, assuming that the authors will add their new results to the paper. ** Original Review ** This paper proposes a meta-learning approach to domain generalization, where the method learns a regularization term that is added to the task learning objective. Once the regularizer has been learned across different domains, the paper proposes to train on all domains using this regularizer. The paper additionally proposes to have a feature network that is shared across domains/tasks to enable more scalable meta-learning. The method provides a small, but significant improvement over simply training a model on all tasks, as well as prior domain generalization methods. The experiments provide additional ablations and visualizations for better understanding the method. Pros: The paper addresses an important problem of the ability to generalize well to new domains given no data in the new domain, and provides an innovative approach based on learning to regularize to the learning process. The idea of learning a regularizer is interesting and relevant beyond the particular domain generalization problem setting addressed in this paper, increasing the potential impact of the paper. The experiments and analysis are thorough. Cons: The empirical benefit provided compared to baseline models is relatively small. The clarity of the paper could be improved, both in terms of notation and how it is presented. Further, the Bayesian interpretation should be further fleshed out for it to be meaningful. There are details of the algorithm that are not included in the current version that would make it difficult to reproduce. [Detailed feedback below] *More detailed comments & feedback* Related work: The idea of sharing a feature network across domains has been recently proposed in the meta-learning literature [2]. It would be good to reference [2] in the paper, for the sake of keeping track of relevant work. But, I do not penalize the authors because [2] is not published and came out ~3 months before the deadline. The related work has a detailed discussion of how this method relates to MLDG, but does not discuss the relation to other works. It would be helpful to include such discussion rather than simply summarizing prior methods. Bayesian interpretation: The Bayesian interpretation doesn't completely make sense. First, what is the graphical model underlying this model? It would be helpful to have a Figure that illustrates the graphical model. Is it the same as the graphical model in [1]? [If so, it might be nice to reconcile notation with that prior work, with theta as the prior parameters and phi as the task parameters.] Second, why does a point estimate for phi make sense? How does the computation of this point estimate relate to the Bayesian interpretation? Prior work [1] developed a Bayesian interpretation of MAML using a MAP estimate to approximate an integral, drawing upon a previous connection between gradient descent and MAP inference. A MAP estimate provides a crude, but somewhat sensible approximation, given that the point has high probability. But, in this case, no such reasoning seems to exist. Discussing the connection to [1] would also be useful. Comments on Clarity: 1. An algorithm for what happens at meta-test time would be really helpful. How are the task parameters theta_g initialized? (randomly or using the existing theta_i's?). How many gradient steps are taken at test time? Is it the same as l? Is both psi and theta updated, or just theta? 2. It would be helpful to have a diagram that illustrates the entire meta-learning approach. Figure 1b is a step in the right direction, but it would be nice to visualize the update to phi and to include multiple domains/thetas. The caption of Figure 1 is thorough, which is very nice. But, it doesn't describe how phi is used. Algorithmic+Experimental suggestions and questions: 1. Right now, pairs of tasks are selected for training the regularizer. Since, at meta-test time, the model will be trained with the regularizer across multiple domains, it seems like it might make more sense to have meta-training of phi mimic this setting more closely, having each theta_i being trained across a set of domains, e.g. all but one domain, rather than an individual domain, and then training phi on domains not in that set. Right now, phi is trained only for thetas that were trained on an individual domain rather than multiple domains. 2. Would it make sense to have a replay memory of previous thetas? It seems like phi might learn to be a good regularizer for parameters that have been trained on a task for a long time (and might forget to be a good at regularizing randomly initialized parameters). 3. What are the implications of freezing the feature network? An ablation with varying amounts of layers in the feature network vs. task network could help study this question empirically. 4. In the experiments, it would also be useful to plot phi_i. 5. Are there experimental settings where we would expect more improvement? Perhaps in settings with more domains? More minor comments: - a more common notation for a set of parameter vectors is to use uppercase \Theta (vs. \tilde{\theta}). - A lot of notation is definied in section 3.1 and some of it is not used later, such as M, T, \tilde{\theta}, \mathcal{D}. It might be better to consolidate some of this notation. - "generalization to a new domain is not guaranteed. To accomplish this, we propose...": This wording suggests that the proposed approach guarantees generalization to new domains, but theoretical guarantees are not discussed. - The last paragraph of section 3.1 is confusing because it isn't clear if there is one theta or multiple theta_i. When referring to theta, does the paper mean {theta_1, ..., theta_p} or an individual theta_i or theta_g? - In equation 4 and elsewhere, it would be helpful to put brackets around everything that is inside the gradient operator, to make it clear that the regularizer term is inside the gradient operator. [1] Grant et al. ICLR 2018 [2] Zhou et al. http://arxiv.org/abs/1802.03596

Reviewer 3



This paper trains a parameterized regularizer for a deep net to encourage transfer among multiple domains. The MTL architecture uses hard parameter sharing of a lower-level feature network coupled with task-specific networks in later layers; the regularizer affects the training of the task-specific networks. Overall I found this idea to be simple yet compelling. The authors do a nice job of developing the idea and the paper is, for the most part, clear. The experimental results seem convincing. I do have the following questions: 1.) The method is training the meta-regularizer in a batch fashion, but how dependent is the process on all of the tasks being available up front? Is it possible to add tasks and still have the regularizer be effective? (I suspect the answer depends a great deal on issue #2 below.) Also note that this question does go outside of the scope of this paper, but would be good to think about for the future. 2.) The major question that is left open is how similar the domains must be in order for this method to work effectively. I would imagine that the domains must be rather similar, but the paper does not examine this aspect, and so it remains a major unanswered question of the paper, and one that truly needs to be addressed in order for this to be a complete examination of this topic. 3.) Section 3.3 seems to imply that you first train all of the task networks, then the meta-regularizer, and then retrain the task networks with the final meta-regularizer. But these steps could all be interleaved, with the meta-regularizer affecting the task networks, and vice versa. Was this explored, or are you really just taking one step along the meta-regularizer - task nets optimization path? Questions 2 and 3 are the major open issues with this work, while Q1 is much more mild and more forward-looking than a criticism.

Reviewer 4



Summary: This paper addresses the topical problem of domain generalisation. Given multiple source datasets for training, it aims to meta-learn a regulariser that helps the model generalise to unseen domains. More specifically, the regularizer to learn is a parameterised function, of which a simple form can be \sum (\phi_i * \theta_i), where \theta is the classifier-model parameter, and \phi is the regularizer-model (or meta-model) parameter. In this way, the regularizer learns to punish the model complexity in order to improve the performance on a held-out domain (domain-(b) in author’s notation) after several SGD updates based on domain-(a), so the trained model is expected to generalize well on unseen domain. Results show good DG performance on PACS object recognition and amazon review sentiment-classification benchmarks. Strengths: + A technically sound method for achieving DG, with a well motivated, and intuitive underlying mechanism. + The key advantage of the proposed method is that, since the meta-learning part happens in regularization terms, the updating for classifier model is not limited to one-step only. In contrast to [8,20], such multi-step updates will lead to higher-order gradients. + The design of the shared feature+task-specific networks improves scalability. (However it is simple to realise the same modification for [8,20].) + Good that positive results are obtained on two very distinct kinds problems of image and text. And also with distinct architectures (AlexNet, ResNet-18/50, MLPs). + Analysis of weights in Fig2 is interesting. Weaknesses: 0. Novelty: While the motivation is quite different, the key schema of this submission is still quite close to [8]/[20], i.e., in each iteration, two domains ((a) and (b)) are sampled as meta-train (train set) and meta-test (validation set). This somewhat caps the novelty. 1. Presentation/Explanation: — Eq(2) seems like un-necessary “Mathiness”. Makes it look fancily Bayesian, but then the paper doesn’t then proceed to actually do anything Bayesian. Sec 3.2 could be removed without affecting the contribution. — The key details of the method is explained in L134-168. However particularly Sec 3.3 L139-168 are rather densely/unclearly written and very hard to parse. If I wasn’t very familiar with this topic and methodology, I would have little hope to understand what is going on. Fig 1(b) is too vague to be of any help deciphering this paragraph. Better to spend the extra space of Sec 3.2/Eq 2 explaining everything better. 2. Meta-Train/Train mismatch: Meta-learning setups normally strive to mimic the way in which the method will be used in practice. But here it seems there is a mis-alignment of train/test conditions for episodic meta-learning. The meta-learning is trained “1-1”: working on pairs of domains (a,b) \in Source Domains by training on domain a, and evaluating on domain b. However, the way in which the regulariser is actually used is “Many-1”, training on *all* the sources. IE: the Meta-Reg parameters are tuned to work with much less data than they are actually exposed to when used in practice. This may lead to sub-optimal outcomes. 3. Competitors: - It would have been good to have some other DG baselines to put the results, particularly on the sentiment dataset, in context. EG: Domain-adversarial, [A] CrossGrad (ICLR’18), [B] CCSA, Doretto, ICCV’17, [C] DICA, Muandet, ICML’13. - For Tab4, it would be helpful to have a baseline with conventional (\phi_i=constant) l1 regularisation. Randomly weighted l1 seems like a weird, and possibly bad baseline. 4. Intuition. While the method is intuitive and appealing at a high level, it would be nice with some more analysis about exactly how it works (i.e. beyond Fig 2), and what are the assumptions it needs. - EG: How works: For example is its effect via reducing overfitting per-se, or via leading the optimizer to find an equally good or better fit that is a more robust minima? (EG: Something like worse train loss vs same or better train loss compared to baseline?). - EG: Assumptions: Are there any assumptions? For example does it only make sense if all train+test domains used are independently sampled from the same distribution over domains, or is no such assumption necessary? Could one imagine a specific group of source vs target domains that are different in some (possibly pathological) way such that this method definitely goes wrong and exhibits "negative transfer" to the new domain? ( Providing a synthetic example to break it should not be a reason to reject, but rather a positive to clarify to readers the conditions under which the method is/isn’t expected to work). 5. Related work. Although this paper is focused on DG. The mechanism is to learn a regularisation hyper parameter. It would therefore be good to explain how it relates to the growing body of related work on gradient-based hyper-parameter learning so readers can understand the connections. EG: [Gradient-based hyperparameter optimization through reversible learning. ICML’15; Forward and Reverse Gradient-Based Hyperparameter Optimization, NIPS’17]. Other minor: - L155: “It is important to note that dependence of \Phi of \theta comes from gradient steps in Eq 4”. Should it say dependence of \theta on \phi? Typo: Line 158, pictoral -> pictorial Fig.2 Description: regularizaton/ regualrizer -> regularization/regularizer Line 249, intial -> initial Fig 1 (b) “l steps” is visually similar to “1 steps”, maybe authors can use another symbol.